# The COVID-19 Pandemic Response and Its Impact on Post-Corona Health Emergency and Disaster Risk Management in Iran

Nader Ghotbi

Graduate School of Asia Pacific Studies, Ritsumeikan Asia Pacific University, 1-1 Jumonjibaru,
Beppu City 874-8577, Oita, Japan; nader@apu.ac.jp

**Abstract:** This paper examines the COVID-19 pandemic response in Iran and offers speculations on the possible impact of its experience on the future response to other health emergencies and disaster risk management based on the lessons learned. The COVID-19 experience in Iran is unique in several aspects, including the significant role played by the healthcare workers' sharing and exchange of information through Internet-based networking applications, and a sociocultural environment that was weakening public trust and cooperation in the use of preventive strategies such as less than the optimum wearing of face masks and attending large social gatherings. There was also hesitation in receiving the necessary vaccine doses due to public skepticism over the effectiveness of domestic COVID-19 vaccines. Furthermore, healthcare workers and health services were afflicted with a lack of sufficient manpower and material resources to fight the pandemic. Moreover, a strong and mostly negative influence of political agenda and religious influence on preventive health policies, especially an initial governmental ban on the import and use of Western vaccines and the pressure to hold religious festivals during the outbreaks, were prevalent. The lessons that can be learned from this ongoing crisis include the value of independent healthcare information networks, transparency in the communication of health information to the public to get their trust and cooperation, and an emphasis on the separation of health policies from political and religious interference.

**Keywords:** COVID-19; health emergency; healthcare information; Iran; public health

## 1. Introduction

The COVID-19 outbreak began in Iran in the city of Qom after the first deaths associated with COVID-19 were detected on 19 February 2020 [1,2]. Iran was one of the countries that were hit hard by the pandemic, and many aspects of life were severely impacted by the spread of the infection and the large load of patients in need of treatment. However, to understand the huge burden of COVID-19 in Iran, it may be better to refer to the latest data on the number of infections as well as the number of vaccinations as one of the main responses of the healthcare system to severely contagious infections. After almost 2 years and 7 months, by 27 September 2022, the official data released by the Iranian Ministry of Health and Medical Education reported 7,551,022 total diagnosed cases based on 54,164,524 conducted tests with 7,327,349 patients recovered, 141,471 total fatalities, and 154,840,917 vaccinations performed in total (65,093,835 first dose, 58,495,796 s dose, and 31,251,286 third dose).

In August 2022, Iran was experiencing the seventh wave of the COVID-19 outbreak, while back in February 2022, Iran had gone through the sixth wave of the COVID-19 outbreaks and, at its peak, was registering a tally of about 150,000 new daily infections (Figure 1). The number of daily fatalities during the sixth wave was estimated at 150 to 190, which was higher than the fifth wave. This may be related to a relatively late referral to healthcare centers as well as inadequate treatment because of the large load of patients in need of healthcare. Almost one-third of the diagnostic tests performed in February 2022 returned positive. One of the unique characteristics of the sixth wave was the large

increase in the number of infections among children, who accounted for 20% of all new cases. The number of new daily hospital admissions because of severe COVID-19 passed 2000. However, schools were allowed to close or switch to online teaching only in areas with the highest infection rates.

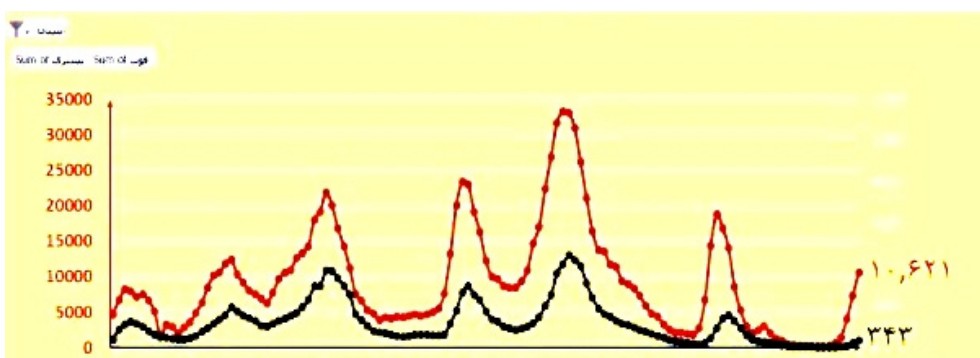

**Figure 1.** Official weekly reports of the number of admissions (red line) and mortality (black line) from the start of the COVID-19 epidemic in Iran till week 128 (5 August 2022). The seven waves of COVID-19 infection in Iran can be seen from the first wave (far left) to the seventh (far right) on the curve. The vertical axis at the left measures both the number of daily admissions due to COVID-19 in red (ending in 10,621 admissions on the 128th week) and the number of daily fatalities in black (ending in 343 admissions on the 128th week). The X-axis depicts the time from the 1st to the 128th week of the COVID-19 epidemic.

Literature review through a search for information in formal publications, including government reports and academic journals, can play a limited role in studying the response to COVID-19 in Iran. On the one hand, Shamsi et al. identified 849 research documents on the topic of COVID-19 from 3450 Iranian researchers, cited by WOS, PubMed, and Scopus, with an average citation per document of 2.2 [3]. On the other hand, San et al. explained how the loss of transparency in authoritarian governments and their attempt to present themselves as successful might lead to a so-called "authoritarian advantage" in dealing with crises [4]. In the case of Iran, several published papers directly refer to the important contributions from government agencies and instead blame "*The US unilateral economic sanctions against Iran*" while citing an article that refers to petrochemical sanctions [1]. Pourghaznein and Salati claim: "*A new trend in fighting the disease has begun. In this trend, different groups of people like clerics and cleric students, Basij, medical and non-medical students have taken measures in fighting the Corona Virus*" [5]. These examples demonstrate the difficulty of walking on a safe line between discussing factors responsible for crisis mismanagement while trying not to cause displeasure to government agencies, especially if the authors hold an official position or are teaching in a medical school. Therefore, getting to the truth of the matter through literature review may not be an easy task.

However, some of the literature has indirectly referred to the underlying problem. For example, Zarei et al. cautiously conclude: "*people expect the government and other responsible institutions to minimize the burden of this pandemic through adopting effective policies*" and "*to become aware of the expectations of people and develop better strategies*" [6]. They also suggest that "*authorities and news agencies should observe the [principles] of honesty and transparency*". For instance, the real number of COVID-19 fatalities in Iran has been estimated to be at least three to five times larger than the official data of 141,471 total deaths, and unofficial estimates of the total number of COVID-19 casualties in Iran are in the range of 400 to 500 thousand people. Many of the deaths due to COVID-19 have been attributed to general causes such as cardiorespiratory arrest and thus not registered as caused by COVID-19.

This study attempts to answer a number of questions raised by the World Health Organization (WHO) regarding the emergency response to the COVID-19 outbreak and its possible impact on post-Corona health emergency and disaster risk management in Iran. The research questions include an inquiry into the difficulties and challenges experienced during COVID-19 in terms of human resources, health service delivery, logistics, the responses to these challenges and the lessons learned from the COVID-19 pandemic to prepare for the future, and the influence of the challenges and responses on the present and post-corona Health-EDRM system.

## 2. Materials and Methods

This paper gets most of its information from voluntary groups of medical doctors and other healthcare workers directly engaged in diagnosing, treating, and other necessary care for COVID-19 infection in Iran. Internet messaging applications that may be installed on mobile phones are widely popular among all societal groups, including medical doctors and other healthcare workers. Among these, Telegram (https://telegram.org) has especially been used to set up several data sharing and information distribution channels and discussion groups with the participation of thousands of medical doctors and other healthcare workers dealing with the COVID-19 outbreak in Iran. The associated Telegram channels and groups contain a lot of gathered information that is checked by the admins and some other members, discussed by the experts in the group, and made available for further scrutiny and debate by the whole group. Moreover, a lot of information collected from English language journals and news agencies is translated into Persian so that it can be used by the Iranian healthcare community. The Telegram groups used for access to the needed information to write this paper include the following: COVID-19 Updates (more than 4500 subscribers), COVID-19 Research Group Iranian Healthcare Personnel (more than 3000 subscribers), Scientific Collaboration on Inpatient COVID-19 Cases (group name translated from Persian, with more than 1900 subscribers), Afflicted by Corona-Legal issues (group name translated from Persian, with more than 1500 subscribers), Scientific Collaboration on Scientific Papers (group name translated from Persian, with more than 800 subscribers), as well as a few groups set up on the WhatsApp messenger (https://www.whatsapp.com). The channels and groups are managed on a daily basis by the so-called admins that check the credibility and relevance of the material to help prevent the spread of false and misleading information. However, because of the Internet use limitations set up by the government, access to Telegram requires the use of VPN and Proxy codes. Therefore, many users have had difficulty using the shared information and, in some cases, switched to the Whatsapp and Instagram Apps, which were less affected by those limitations. Nevertheless, the Telegram channels are still of the highest quality and updated with the latest data (see Figure 2). Moreover, since October 2022, the Iranian government has indefinitely decided to disable connection to Whatsapp and Instagram Apps.

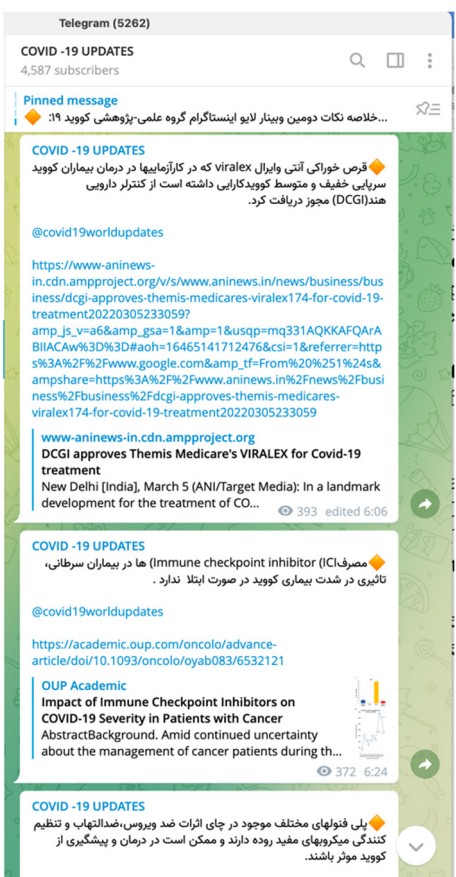

**Figure 2.** This screenshot from a Telegram channel called COVID-19 Updates demonstrates how new medical information released in English is being partly translated to Persian with links to the Internet source provided for 4587 subscribers. This channel is used to present updated medical information about all aspects of COVID-19, including treatment methods, medicines, etc. Other Telegram groups allow discussion by the members and their exchange of data, information, and personal experience regarding COVID-19 management and healthcare issues.

## 3. Results

This section first explains the issues related to the management of human resources in response to the outbreak of COVID-19 in Iran. Then the delivery of health services and, finally, the relevant logistics will be discussed. There were many challenges related to the human resources needed during COVID-19. A primary challenge was protecting the healthcare workers from the infection brought to them by patients. Unfortunately, many healthcare workers got infected by their patients, and some of them died. By 25 July 2022, 699 healthcare personnel in Iran, including medical doctors, nurses, midwives, and other healthcare-related staff, have been reported deceased because of COVID-19 infection. The other challenge faced by the healthcare community was anxiety and stress over the high risk of infection through the provision of health services to the patients. Zandian et al. found high levels of stress among nurses taking care of COVID-19 patients in Iran [7]. Shahriarirad et al. identified the healthcare occupation as a factor increasing the risk of depression and mental health breakdown. They attributed it to "*the inadequacy of protective equipment and the stretching of hospitals to the breaking point due to the rapidly increasing number of patients seeking medical treatment*" [8]. A daily newspaper in Iran had a worrying report of the exhaustion of the healthcare personnel and the large number of medical doctors who either leave the country to look for jobs overseas, quit working in the healthcare system, suspend working in a clinical capacity or in the public sector, discontinue medical education and specialization and otherwise leave the service (Figure 3).

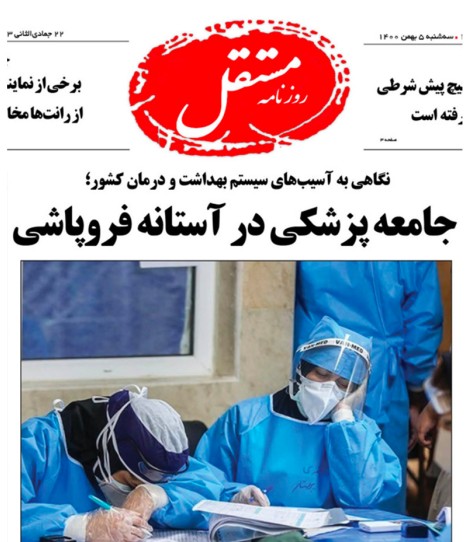

**Figure 3.** This image from an Iranian daily newspaper called "Independent Daily" (in Persian), dated 25 January 2022, has its main article at the center of the page titled: "Medical community at the edge of collapse, a look at the problems in the healthcare system". The photo that follows the main title shows two exhausted medical doctors at work. The article describes the aftermath of the exhaustion of the healthcare personnel as: leaving the country, leaving the medical occupation, leaving the patient treatment role, leaving the public sector of healthcare system, loss of interest in continuing medical education, loss of interest and withdrawing from positions in medical education, and severe loss of manpower in basic and primary healthcare.

Jam-e Jam Daily announced that other than human resource casualties among the healthcare personnel, more than 800 taxi drivers and 120 teachers had died because of COVID-19, more than 612 thousand people moved below the poverty line, and 210,000 and 760,000 students dropped out of the primary school and high school system, respectively. An explosion in mental health problems tripled the number of requests for consultation, with a 77% increase in reports of domestic violence. Parvar et al. found an association between higher age and perceived stress, while the male gender and higher income were found to be associated with higher resilience scores [9]. Yoosefi Lebni et al. refer to the economic, social, cultural, legal, and political factors that had an impact on the response to the COVID-19 pandemic in Iran [10]. In the section on political factors, they mentioned the contradictory messages from the authorities regarding attendance at religious rituals, which they blame for the rapid spread of the infection throughout Iran. They also explained how patients with a CT diagnosis of COVID-19, albeit with no laboratory test conducted, would not be included in official reports.

In response to these challenges, unfortunately, many healthcare workers have given up their job temporarily or for the time being because of the unbearably large load of work, increasing risk of treating (potentially) infected patients, suffering from severe COVID-19 symptoms after getting infected, and disappointing economic prospect related to staggering inflation and increasing costs of operation while medical fees are relatively controlled by official directives. Some have already migrated or are seriously considering emigration to other countries, such as Canada, and some have entered other professions using their savings as an investment. According to government data, about 4000 medical doctors are considering migration in 2022. Many specialists are moving to nearby countries such as Oman in search of a good income and better working conditions. Nurses are also looking for countries that welcome immigrants with their profession, especially after many of them were laid off after the sixth wave of the COVID-19 epidemic was over, partly to manage the dwindling budgets of healthcare services in Iran.

To help lower the workload and its associated pressure on the healthcare staff, medical universities tried to transfer a larger load of work to medical interns and residents who

have less control over the working hours and duties. This has resulted in serious health problems, and, in a few cases, led to the suicide among some of the severely stressed or caused the death due to exhaustion of some others. Another strategy to control the increased workload of hospitals due to COVID-19 was to limit and postpone elective and nonemergency operations and the treatment of less serious illnesses. The Ministry of Health also issued orders to require physicians' compulsory attendance at work, including private practices. On the other hand, some medical doctors and healthcare workers have sacrificed their own health and welfare to serve patients in need of care. Vahidi et al. have referred to the role of citizen science volunteers who supported the government data collection and contact tracing, promoted the use of face masks to help reduce the spread of the infection, and helped increase the awareness of the public regarding COVID-19 through the use of social media platforms [11].

Zarei et al. used online surveys to study the expectations of the public from the government and its response to the COVID-19 outbreak and came up with three major areas: health-related expectations, policy-related expectations, and mass media-related expectations [6]. Based on the results of their study, most respondents to the survey wanted the government to provide support and follow-up of the patients and their families, including both health-related and financial support, and conduct proper isolation, restriction, and monitoring of the infected and be more honest and transparent about the COVID-19 situation [6]. However, the healthcare manpower is already exhausted, and many other healthcare resources have been used up. There may be a surge in migration out of Iran if a wide-scale and serious health emergency/disaster happens. As described earlier, the most recent government data in 2022 suggests that about 4000 medical doctors are looking to migrate to other countries.

Abounoori et al. used factor analysis to develop and validate a knowledge and attitude scaling tool regarding the COVID-19 outbreak among the Iranian population and came up with two factors for health literacy and home health empowerment [12]. They recommend using such a scale to monitor the knowledge and attitude of the public to help inform policymakers and healthcare providers of future transmission risks. The challenges imposed by the COVID-19 pandemic in Iran will have an enduring impact on future responses to a possible health emergency or disaster, and the challenges and responses to the present epidemic could have an impact on the post-Corona Health-EDRM system. Gharebaghi and Heidary explained how directing the national task force against COVID-19 was assigned to the Minister of Health and Medical Education, and high-level provincial committees chaired by the governor and the chancellor of the medical university were established. Then various committees were formed to enhance intersectoral collaboration [13]. Amiri et al. examined the role of the National Headquarters for the Control of COVID-19 Epidemic in response to the COVID-19 outbreak, such as the establishment of several scientific committees, including the COVID-19 National Epidemiology Committee. They reviewed the Epidemiology Committee's missions, structures, achievements, and challenges of the [14]. Their study suggests that the committee's working groups may have improved "*data registration/usage, provincial data quality at provincial levels, and perception of the epidemic situation in the provinces*". The committees also helped elaborate the control policies in the changing stages of the epidemic through data analysis, epidemiologic investigation, and mathematical modeling. Therefore, they concluded that the structure and experience gained by the committee could be used in similar situations in the future and recommended: "*effective interaction, collaboration, and data flow between the committee and a broad range of organizations within and outside the Ministry of Health and Medical Education*" [14].

On the other hand, the experience gathered through the response to the current pandemic may help avoid some of the mistakes that were made. For example, the value of networking and exchanging information through new technology and Internet-based applications has been made clear. There is hope that there will be less cover-up and more transparency in strategic decisions, such as purchasing vaccines approved by qualified institutes. There is also hope that state politics will not try to influence healthcare responses or

create distortions in valuable data and information. An important consideration is building up trust among the public so that next time they will follow the experts' recommendations on controlling the health emergency situation.

The next category of issues related to the response to the outbreak of COVID-19 in Iran is the management of health service delivery. One of the challenges experienced during COVID-19 was the lack of strict observation of the protocols to prevent the spread of infection by the public, which has been one of the main difficulties faced by the healthcare service. Some of the most common manifestations of this problem include a denial by patients suffering from coronavirus symptoms saying that it must be a common cold or allergy and discontinuing isolation and quarantine too soon, claiming their symptoms disappeared and they must have recovered. Another problem was patients attending gatherings while infected, claiming that wearing a mask would be enough to prevent the spread of infection and claiming they must be immune to COVID-19 because they have not been infected so far. Additionally, some patients avoided the vaccine claiming that it would not help and could be dangerous. Moreover, some patients were not seeing a doctor in the early phase of infection, waiting until it was too severe and commonly too late for the treatment to have a strong effect on the course of the infection.

In the early stages, the medical personnel faced a severe shortage of protective equipment, and there were insufficient care facilities for the severely affected patients. People rushed to the stores to purchase masks and disinfectants, which created a black market and made it more difficult for hospitals to acquire the protective equipment they needed [1]. There was a lack of coordination between the government and the medical community regarding the need for preventive measures. This lack of coordination has continued to impact the response to the pandemic negatively. For example, the Ministry of Higher Education and the Ministry of Education were once insisting that school classes should be held on-site. In contrast, the medical community was responding to the sixth wave of infection and wanted the schools to delay students' return to classes and to continue with online education until the daily load of new infections fell to a manageable level. However, to spread awareness about the risk of infection in different areas, a color-coding system with red, orange, yellow, and blue colors has been used to warn the public about the community level of infection spread (see Figure 4).

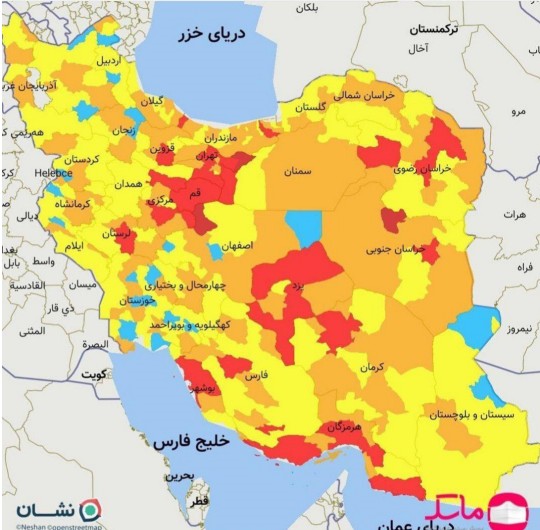

**Figure 4.** This map of Iran uses a color code system of blue, yellow, orange, and red to show the relative level of risk of infection with the new coronavirus in various prefectures at the time of publication, based on the number of daily new infections registered in hospitals. The use of such maps provided an easy means of infection risk communication to the Iranian community, especially during periods in which many people consider traveling to a domestic destination.

One of the issues causing much confusion was the value of hydroxychloroquine in prevention. There were so many studies for and against it, and the debate continues. However, it showed the limitation of available options and how even a small chance of effectiveness caused thousands of medical doctors and other healthcare workers to take a chance and prescribe it. Another example was the use of drugs containing opium by people based on a rumor that drug addicts were somehow immune from COVID-19 [15].

In response to these challenges, the Ministry of Health and Medical Education has released many guidelines in the Persian language for the outpatient and inpatient diagnosis and treatment of COVID-19, with the 11th edition dated January 2022. These guidelines were prepared by the "national scientific committee to manage COVID-19" with 12 medical authors and more than 50 medical experts in its list of experts. The 11th edition is 106 pages and covers all practical aspects of dealing with clinical decision-making, including the epidemiological definition of cases, course of infection, indications for outpatient treatment, referral to hospital for admission, symptomatic and lab screening, isolation methods, preventive strategies including vaccination, mental health support, and other issues. They recommended a cocktail of therapeutic agents for relatively severe outpatient cases, which includes doxycycline, Montelukast, vitamin B12, bromhexine, famotidine, and magnesium. In more severe cases, Sofosbuvir (for Omicron), Paxlovid (nirmatrelvir and ritonavir), remdesivir, molnupiravir, and a low dose of corticosteroids were recommended when available. There is a need for an information network of healthcare workers dealing with COVID-19 to share the latest knowledge about available therapies, facilities where patients may be admitted for intensive care, where vaccines can be administered, etc. Currently, informal and voluntary networks exist, but they face the problem of government bans and social media filtering, including Internet-based applications used by healthcare workers. Nevertheless, they have been able to use strategies such as VPN and proxy servers to overcome these barriers and continue networking in response to the pandemic.

The third category of issues related to the response to the outbreak of COVID-19 in Iran is the management of the logistics needed. One of the essential supplies to help control COPVID-19 was the provision of vaccines approved for use against infection with the new coronavirus. The grave experience of multiple failures in the supply of approved vaccines shows that politicians should stay away from making healthcare decisions that are not within their expertise. For example, the supreme leader had made a speech in which he doubted the quality of American- and British-made vaccines. This resulted in mainly purchasing Russian- and Chinese-manufactured vaccines known as Sputnik V, Sinovac, and Sinopharm. This happened while most countries were putting orders to purchase Pfizer, Moderna, and AstraZeneca vaccines. Although the decision of the supreme leader was taken back a few months later, and at least AstraZeneca vaccines were purchased and distributed for use, the bleak aftermath of that decision had already resulted in many deaths because of the long delay in the purchase and an apparent ineffectiveness of some of the purchased vaccines. Moreover, the purchase and supply of the Russian and Chinese vaccines were accompanied by several mistakes, and the hearsay about some lower-quality batches of vaccines or fake ones filled with normal saline and sold by fraudulent individuals damaged the public trust in vaccination. In one case, the World Health Organization (WHO) had to release a warning on its homepage regarding the fake quality of some vaccines distributed in Iran [16]. Even though there were so many national vaccine projects ongoing at the same time, a number of technical, logistical, and managerial issues in the manufacturing process reduced the supply of even the most well-known brands to levels that were not enough to cover the need when the vaccination campaign started. Therefore, millions of vaccine doses were imported from China and Russia, as well as the AstraZeneca vaccine. This happened while a huge budget was allocated to the domestic production of COVID-19 vaccines, and 14 such projects were financed. Although these projects claimed success in producing an effective vaccine, almost none of them had completed the phase 3 trials, and many were not even in phase 2. Therefore, there was

no international approval of their safety and efficacy. This caused many people to avoid receiving domestically branded vaccines.

A million doses of one of these vaccines, called Barekat, was recently donated to the African country of Mali because the domestic demand was much lower than the manufactured doses. Barekat was the first Iranian COVID-19 vaccine to receive authorization for use on 14 June 2021. Pastu Covac, called Soberana 2 in Cuba, was another vaccine developed jointly by the Pasteur Institute of Iran and Cuba's Finlay Vaccine Institute. It was especially recommended for use in children because of the claimed safety. Razi Cov Pars was the second domestically produced COVID-19 vaccine that reached the phase of clinical trials but was given emergency use authorization in Iran on 31 October 2021. However, the company announced on 6 July 2022 that it was closing its manufacture of the vaccines because of a lack of demand, and 3.5 million vaccine doses were being stored.

The Iranian Minister of Health announced on 22 February 2022 that about 90% of the COVID-19 deaths in Iran had occurred to those who were not vaccinated. At the same interview, he said that 820,000 AstraZeneca vaccines, which were imported from Poland, but the point of origin was the US, would be returned to Poland, and "approved" vaccines would replace them instead. The issue of which vaccines are more effective has unfortunately been politicized, and the medical and other healthcare personnel have worries over the quality and effectiveness of vaccines to which they have had access. There were many reports of low-quality cheap masks imported from overseas that are not useful in prevention. Surgical masks are no longer enough to protect healthcare staff from infection with the new coronavirus. Instead the use of N95 masks is being promoted, especially among those working in the health services system.

A major issue was the decision of where to purchase vaccines. As mentioned before, the supreme religious leader, early in the course of the response to the pandemic, suggested that vaccines made by the US and the UK should not be trusted as the data from these countries showed a large number of infections and, thus, the vaccines cannot be effective. Following that comment, the government purchased vaccines from Russia and China while the Russian vaccine had issues that would not let the WHO approve it for preventing COVID-19. The best vaccines were, in fact, mRNA vaccines, such as Moderna and Pfizer, that had been banned following the commentary of the supreme leader. Many healthcare workers had no choice other than to accept the vaccines that had been purchased. The reports that followed demonstrated that these vaccines were not as effective as mRNA vaccines; several medical doctors got infected and passed away though they had received the government-approved vaccines.

In the meantime, the government financed research on developing up to 14 different vaccines, and they reported good results for most of these experimental vaccines. However, it was later revealed that mass production of the vaccines was a different matter, and the government was not able to manufacture the needed number of vaccines. A vaccine brand that was released as manufactured by Iran under the name of Barekat got a bad reputation, and rumors about its authorization process caused many people to lose their trust in COVID-19 vaccines. Meanwhile, Iran received millions of doses of the AstraZeneca vaccine under pressure and protest by the medical community and started administering them. However, some fake vials were soon discovered, and WHO released a statement of warning over these incidents. The released warning by the WHO said: "*This WHO Medical Product Alert refers to falsified COVID-19 VACCINE AstraZeneca (ChAdOx1-S [recombinant]) identified in the Islamic Republic of Iran and reported to WHO in October 2021. The genuine manufacturer of COVID-19 vaccine AstraZeneca (ChAdOx1-S [recombinant]) has indicated that the product is falsified. The falsified product was reported at the patient level outside authorized and regulated supply chains and authorized immunization programmes in the Islamic Republic of Iran*" [16].

Iran mainly used the following vaccine brands for its national immunization against COVID-19: Sinopharm, COViran Barekat, Bharat, Pastu Covac, AstraZeneca, Sputnik V, and SpikoGen. Babaee et al. studied the three most commonly used vaccines in Iran

and reported that Sputnik V caused more side effects than AstraZeneca and Sinopharm vaccines [17]. More recent data released by the government suggest that 90% of 12- to 18-year-old adolescents have been vaccinated, and families have been encouraged to vaccinate their 5- to 12-year-old children. Otherwise, there would be a 20% risk of serious illness among children infected with Omicron. The development of the so-called "long Covid" among a subgroup of patients is especially of concern. Heidari et al. have written about the dire consequences of a significant delay in vaccination administration, *"the collective overwhelming fallacy toward immunization, the polypharmacy controversy, inadequate community-based participation in risk reduction, and noticeable decrease in the public's resilience"* [18].

More recently, the government has been trying to persuade people to get vaccinated especially using domestically produced vaccines, even by setting new regulations. For example, businesses that require a license to work may be forced to close if the permit holder is not vaccinated. In the meantime, the official period of rest from governmental jobs for recovery has been increased from 5 days to 7 days. This period may appear insufficient, but the fact is that many people in the private sector do not take any days off if they are able to go to work and may simply claim they have caught "the common cold". The Ministry of Health and Medical Education had recommended the 5-day period based on their estimate of a shorter duration in the case of Omicron which is the predominant variety in the sixth wave of the COVID-19 pandemic in Iran. The worsening situation of the economy for most people has now turned into a more dire problem than the dangers of the pandemic, and for many people, it is much more important to be able to earn an income. It is estimated that in the sixth wave of the pandemic, 80~85% of the cases were caused by Omicron and 15~20% by the Delta variety. This made the preventive efforts more challenging, and the number of newly infected people went up significantly. Thus the number of severe cases increased even though Omicron usually causes a less severe form of COVID-19 than Delta.

Experts believe the main issue in logistics was the lack of a strong ethical board to oversee the general policies, some ineffective decisions that were made, and the weak implementation of other decisions by managers in the Ministry of Health. It is presumed that a powerful ethical board at the top policy-making level could have prevented the political agenda from entering health policies, keeping the healthcare decision-making process independent from both political and financial gains by profit-seeking influences and providing more transparency and, thus, trust among the people. Besides the shortage of personal protective equipment such as masks and gowns, there were the issues about which vaccines would be purchased and whether there would be a mass manufacturing capacity for vaccines researched by various institutes. Moreover, issues on how the first-line healthcare workers would be provided with the most effective vaccines have been of primary significance.

It can only be hoped that the Ministry of Health will prepare to implement many lessons learned from the mistakes that were made in response to the pandemic: letting an airline continue flights between Iran and China in the early days of the pandemic despite the high risks involved, politicizing the healthcare response and using it to attack their political adversaries in the west and please their political partners in Russia and China, letting religious groups continue with their mass rituals and claim that god would protect them, etc. However, it seems that the response will depend on who will be running the government as president, who will be the head of the Ministry of Health and its main high offices, and whether there will be competing political goals to follow by a higher authority or not. On the other hand, the healthcare community has learned to use new technology to access and share information and exchange data and information to support a network of private volunteers who care about one another, their profession, and the people they serve.

## 4. Conclusions

The healthcare community dealing with COVID-19 patients in Iran is severely exhausted, and the issue of burned-out healthcare workers needs formal recognition and systematic remedies. Severely affected patients who need to be admitted face a shortage in

the number of beds available and the exhaustion of the healthcare team under the pressure of the large load of patients, limitations in the needed therapeutic and protective equipment and facilities, having received only controversial vaccines that may not have international approval, and insufficient financial rewards for their sacrifice.

Sociocultural issues are still a main challenge to preventive efforts. These include people ignoring symptoms and ascribing them to the common cold and continuing to mingle with others, delaying a visit to the health centers until the symptoms become too severe, and positively identified cases breaking their isolation as soon as their symptoms start disappearing. Furthermore, issues include people justifying attendance in group activities while infected by simply wearing a mask and not allowing ventilation in many closed spaces because of cold, noise, or the sense of privacy. Moreover, people would claim they are invincible and would not catch COVID-19 because they appear not to have been infected and not getting vaccinated, claiming it is risky or ineffective.

Many of the currently available vaccines in Iran appear not to prevent infection with the Omicron variety, but they appear to limit the progression of illness and reduce symptoms. This may cause an increase in the spread of infection by infected people who do not observe the orders of isolation and claim they only have some common cold symptoms. As the number of children infected with the Omicron is increasing, it is necessary to get a firm policy on how to stop clusters in schools from happening by closing down classes and moving them to online education, vaccinating children with safe and effective vaccines, and supporting schools in the provision of proper healthcare for teachers and other staff. However, the funds seem to be limited, and the growing inflation has severely limited the ability of schools to implement the needed policies sufficiently.

It can only be hoped that the government will use public media to promote trust and respect for medical doctors and other healthcare workers in the general population. At the same time, the Ministry of Health and Medical Education might consider the financial needs of medical doctors, nurses, and other healthcare workers who risk and sacrifice their own health to provide treatment and care to infected patients in need. An important observation in Iran is the reactionary nature of the response to the crises. Authorities barely come up with an effective plan to predict and address potential problems but commonly wait until a problem happens. Even then, the first response is commonly to blame others rather than implement the needed remedy in a short and effective manner. Some of the activities at the Ministry of Health, especially the release of standard instructions through teamwork by their research and academic teams, have been worthwhile and useful. It can only be hoped that their work will continue and that they will continue to release new updates on the standards of managing COVID-19 patients. Meanwhile, there is a need for a special committee to coordinate the responses of other ministries with such efforts. For example, the Ministry of Internal Affairs recently released an order to hold all higher education classes at any level in actual classes and to discontinue online education without consideration of the growing number of Omicron cases in the epidemic. Apparently, concern over the declining quality of education caused this order to be released. However, there were at least 7 million unvaccinated people who should be vaccinated before returning to face-to-face classes on the campuses. The news that the World Bank accepted to provide a 90-million-dollar loan to Iran for COVID-19 control was a positive development.

A post-Corona health-EDRM system has not been developed yet. There are many reasons, including Iran experiencing another (the seventh) wave of infections, with more than 200 deaths on some days. The Ministry of Health started providing the fourth vaccine dose to medical doctors and those at the front line, but the issue of available vaccine choices and their effectiveness has not been resolved yet. The new government has also tried to use the force of law and financial penalties and fines to make the public follow preventive policies such as travel ban, use of masks, getting vaccinated, and so forth. There are still competing political issues, such as the ongoing talks over the nuclear deal between Iran and the west, regional conflicts ongoing in Yemen and Syria that may still be a priority for the authorities, the risk of new conflicts in the region, such as from the east

(Afghanistan under Taliban control), and west (Kurdish groups). One big concern is the possibility of having a strong earthquake which is long overdue based on the historical record. A strong earthquake could cause a major health disaster and might cause a serious catastrophe. The main positive lesson learned so far has been the networking activity of private physicians and other healthcare workers in supporting each other and sharing and exchanging vital information, which will surely grow stronger unless the government tries to further crackdown on Internet connectivity for political reasons.

**Funding:** This study was funded by the World Health Organization Kobe Centre for Health Development (WKC-HEDRM-K21001).

**Institutional Review Board Statement:** Not applicable.

**Informed Consent Statement:** Not applicable.

**Data Availability Statement:** Not applicable.

**Conflicts of Interest:** The author declares that no conflict of interest exists regarding this study.

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
