# Peer review of "The COVID-19 Pandemic Response and Its Impact on Post-Corona Health Emergency and Disaster Risk Management in Iran"

_sustainability, doi:10.3390/su142214858_

Round 1

Reviewer 1 Report

This is an important issue. The author describe the emergency situation and its management during corona and post corona period. They nicely describe the horrific situations. However, the manuscript need to be polished further to be published in such a reputed journal. My concerns are--

Major

1. An introduction introduces the problem to the audience. But in this manuscript, it seems to be results/observations. Please give a straight forward description of the crisis. 

  2. Methodology is unusually long, make it brief. Directly describe how you retrieve the literatures and information.

3. Conclude your manuscript  directly on the basis of your solid results. It is also too long. It should also contain the preparedness about such digester in future.

Structure of the manuscript

Please reshuffle your manuscript following the instructions given bellow-

1. Introduction

2. Methodology

3. Results and Discussion

4. Conclusions  and future directions

Minor

1. In abstract, please include few introductory sentence at the beginning and a concluding remark at the end.

2. Carefully check the language. It would be 'learnt' not 'learned' throughout the text.

Author Response

Thanks for the very useful comments.

I have made extensive revisions following the comments I received.

Hereby I am submitting the revised manuscript with the changes tracked.

Best regards

Reviewer 2 Report

I would like to congratulate the authors for their interesting and informative work.

This is a study investigating the COVID-19 pandemic response in Iran, considering several aspects that are characteristic for the country. The authors discuss the lessons that could be learned from this crisis, including the value of independent healthcare information networks, transparency in communication of health information to the public, and separation of health policies from political and religious interference.

This is an overall well written paper. Here, I have made a few suggestions that (in my opinion) could help improve the overall quality of the manuscript.

The authors may consider clearly stating the aim of the study at the end of the introduction/background.

The authors may consider providing Figure 1 with the annotations written in English.

The authors may consider providing more information, if available, regarding the “admins” of the channels and groups (e.g., professional/medical background, potential roles in public health authorities).

The authors state: “In one case the World Health Organization (WHO) had to release a warning on its homepage regarding the fake quality of some vaccines distributed in Iran.” They may consider providing a reference here.

The authors may consider discussing the framework and impact of potential collaborations with international health organisations in the management of future public health crises.

Author Response

Thank you for the very useful comments.

I have followed on the comments and made extensive revisions as required.

Please observe the attached file which shows the changes made as tracked by the review function in Microsoft Word.

Best regards

Reviewer 3 Report

The article does not follow the journal's guidelines and much of the content is not in English. 

Author Response

Thanks for the useful comments.

I have done extensive revisions to follow on the required changes.

Hereby I am attaching the file that shows the made changes as tracked by the review function in Microsoft Word.

Best regards

Round 2

Reviewer 1 Report

The manuscript has been improved and can be accepted to be published. However, still I recommend to label the Fig 1 and Fig 4 in English, which are the main figures and is now in Persian language. I have deep respect to the native language of the authors but it will be published in an international journal, therefore, the manuscript must be readers friendly. At least, please keep the English   translation in parenthesis. 

Author Response

Dear Reviewer,

Thanks for your comments. I have followed them through and made the required changes.

Best regards

Reviewer 2 Report

Thank you for taking the time to consider my suggestions and comments and revising your manuscript accordingly.

Author Response

Dear Reviewer,

Thank you. I have done some more improvement based on the style of the journal.

Best regards

Reviewer 3 Report

First of all, I would like to congratulate the author for the work, as well as the revisions that have increased the quality of the manuscript. 

However, the paper is still not adapted to the conditions of the journal.

Author Response

Dear Reviewer,

Thank you for your comments. I have done more improvements to the content, and changed the style according to the standard of the Journal. As such, I changed the formatting from WHO required numbers to a format where the 3 sections of results are mentioned inside the text and therefore the Journal format has been followed.

With best regards

Round 3

Reviewer 3 Report

The article is still not in the issue's template.